# Occurrence and Characteristics of Carbapenem-Resistant *Klebsiella pneumoniae* Strains Isolated from Hospitalized Patients in Poland—A Single Centre Study

**DOI:** 10.3390/pathogens11080859

**Published:** 2022-07-29

**Authors:** Jolanta Sarowska, Irena Choroszy-Krol, Agnieszka Jama-Kmiecik, Beata Mączyńska, Sylwia Cholewa, Magdalena Frej-Madrzak

**Affiliations:** 1Department of Basic Sciences, Faculty of Health Sciences, Wroclaw Medical University, Chalubinskiego 4, 50-368 Wroclaw, Poland; jolanta.sarowska@umw.edu.pl (J.S.); irena.choroszy-krol@umw.edu.pl (I.C.-K.); magdalena.frej-madrzak@umw.edu.pl (M.F.-M.); 2Department of Pharmaceutical Microbiology and Parasitology, Faculty of Pharmacy, Medical University, 50-367 Wroclaw, Poland; beata.maczynska@umw.edu.pl; 3Medical Laboratory Synevo, Fieldorfa 2, 50-049 Wroclaw, Poland; cholewas@synevo.pl

**Keywords:** carbapenemase-producing *Enterobacteriaceae*, NDM, MBL, OXA-48, multi-drug-resistant strains

## Abstract

The global emergence and spread of genes responsible for the production of ESBL (extended-spectrum beta-lactamases) and carbapenemases in *Klebsiella pneumoniae* isolates poses a serious threat to public health. The aim of this study was to retrospectively analyze the frequency of occurrence and drug resistance of selected alarm agents isolated from patients of the specialist hospital in Wrocław. A total of 13,528 clinical materials collected from patients of a specialist hospital in Wrocław were analyzed in the period from 1 January 2020 to 31 December 2020. Overall, 3894 bacterial strains were isolated from clinical materials, including 416 *K. pneumoniae* isolates. *K. pneumoniae* that showed resistance to ETP (ertapenem) and/or MEM (meropenem) were tested using phenotypic tests for the detection of KPC (carbapenemase-producing *Klebsiella*), MBL (metallo-β-lactamase) and OXA-48 (oxacilinase-48) carbapenemases. In the case of a positive or doubtful result of the phenotypic test, immunochromatographic tests and the CarbaNP test were performed. In total, 58 *K. pneumoniae* isolates resistant to 1 or more carbapenem antibiotics were isolated. Of the 58 strains, 16 (27.6%) were isolated from rectal swabs conducted on CPE (carbapenemase-producing *Enterobacteriaceae*) carriers. In the case of CRE (carbapenem-resistant *Enterobacteriaceae*) *K. pneumoniae*, carbapenemases were detected in 28/58 (48.3%) isolates. Notably, 23/28 *K. pneumoniae* isolates produced MBL/NDM (New Delhi metallo-β-lactamase) (82.1%), 5/28 produced VIM (Verona-intergon-encoded metallo-β-lactamase) (14.3%), and one produced MBL/NDM + OXA-48. Carbapenemases were detected in 13 of 16 (81.3%) carbapenem-resistant *K. pneumoniae* isolates derived from rectal swabs. The significant participation of CRE and CPE isolates in the infections proves the need to test patients admitted to hospital wards for their status as a CPE carrier in order to limit the emergence of new epidemic outbreaks.

## 1. Introduction

In 1983, β-lactamases of the ESBL (extended-spectrum beta-lactamases) type, enzymes capable of hydrolyzing penicillins, cephalosporins and monobactams, were described for the first time in Germany. Since then, a continuous increase in the number of infections caused by bacteria presenting this resistance mechanism has been observed. In the treatment of infections, whose etiological factors are ESBL (+) bacteria, other groups of antibiotics are used, including carbapenems belonging to β-lactams, which are considered antibiotics of last resort [1].

Carbapenems are closely related to penicillins and cephalosporins, but due to differences in their chemical structure, they constitute a separate group of β-lactam antibiotics. Similar to other antibiotics from this group, they are characterized by bactericidal activity due to high affinity to PBP-2 (penicillin-binding protein), as well as PBP-1, PBP-1B, PBP-3, PBP-4, and PBP-5 [2,3]. Carbapenems are very similar in structure to acylated D-alanyl-D-alanine, which is the terminal amino acid residue of the peptidoglycan. The structural similarity allows these antibiotics to penetrate the bacterial cell wall and irreversibly bind to PBP proteins, interfering with the further synthesis of the cell wall by inhibiting the formation of bridges between peptidoglycan subunits [2,3]. This process prevents the formation of an integral structure and exposes the bacterium to autolytic enzymes with subsequent cell lysis. For this reason, carbapenems and other β-lactam antibiotics work best in acute infections where there are numerous young, dividing bacterial cells [4].

There are currently four carbapenem antibiotics registered in Poland: imipenem (IMP), meropenem (MEM), ertapenem (ETP) and doripenem (DRP). Due to the difference in their action, IMP is mainly used against Gram-positive bacteria, while MEM and ETP work better against Gram-negative bacteria [5]. The exception is *Pseudomonas aeruginosa*, which is resistant to the effects of ETP. Despite similar indications for the use of IMP and MEM, in the case of meningeal infections, only MEM is used because it has a much better penetration into the CSF [2]. Carbapenems are also used, inter alia, in the case of sepsis and fever of unknown etiology, severe hospital and ventilator pneumonia in patients in intensive care units (ICU), skin and soft tissue infections and osteoarthritis [6]. 

Unfortunately, the frequent and unjustified use of this group of drugs has led to the emergence of new resistance mechanisms in the case of *Enterobacteriaceae*. Among them, the most important is the production of carbapenemases, enzymes belonging to the β-lactamase group capable of breaking down carbapenems and other groups of β-lactam antibiotics. Carbapenemase-producing *Enterobacteriaceae* (CPE) are strains that have acquired resistance to most antibiotics, including carbapenems [7]. The literature also uses the term CRE (carbapenemase-resistant *Enterobacteriaceae*), which refers to the resistant phenotype, while CPE refers to the mechanism of carbapenemase production [8]. According to the Ambler classification, carbapenemases are divided into 3 classes of β-lactamases (A, B, D) due to their chemical structure [9]. Class A and D are represented by enzymes that contain a serine residue in their active site, which is why they are called serine β-lactamases (SBLs). The presence of serine leads to the hydrolysis of the β-lactam ring by binding to the acyl residue. Class B enzymes, which include, among others, metallo-β-lactamases (MBLs) in their active centre, unlike other classes, have a zinc ion (Zn^2+^). Unlike serine β-lactamases, MBLs act on antibiotics by direct attack of the hydroxide ion, the formation and maintenance of which depends on the presence of the zinc ion [3]. The activity of SBLs can be inhibited by β-lactamase inhibitors such as clavulanic acid, sulbactam or tazobactam. MBLs are resistant to these inhibitors, but they are deactivated by metal chelators, such as EDTA and dipicolinic acid. Despite their in vitro activity, they are not compounds approved for clinical use due to numerous side effects [10].

The carbapenemase KPC (class A) and the metal-β-lactamases VIM, NDM (MBL, class B) and OXA-48 (class D) are the most common in *Enterobacteriaceae* [11,12]. The occurrence of resistance to carbapenems may also be associated with the presence of efflux pumps and variable expression of porin proteins [8,13]. 

Currently, β-lactamases of the NDM type (New Delhi metallo-β-lactamase) are considered the most important from the clinical and epidemiological point of view. The increasingly frequent detection of other carbapenemases heightens the interest in AMR (antimicrobial resistance) among microbiologists, epidemiologists and doctors [14]. For this reason, units responsible for keeping epidemiological statistics and educational activities aimed at reducing or inhibiting the problem of antibiotic resistance among bacteria have been established all over the world. In Poland, such centres are the National Reference Centre for Antimicrobial Susceptibility (KORLD) and the National Antibiotic Protection Programme [15]. Leading centres in Europe are the European Centre for Disease Prevention and Control (ECDC) and the European Committee on Antimicrobial Susceptibility Testing (EUCAST) [16]. 

The aim of this study was to retrospectively analyse the frequency of occurrence and drug resistance of selected alarm agents isolated from patients of the T. Marciniak Lower Silesian Specialist Hospital in Wrocław in the period from 1 January 2020 to 31 December 2020. In the study, particular attention was paid to the *Enterobacteriaceae* family resistant to carbapenems, essentially *Klebsiella pneumoniae*, producing β-lactamases with an extended substrate spectrum (e.g., KPC, NDM, VIM, and OXA-48).

## 2. Results

In the period between 1 January 2020 and 31 December 2020, the following 13,528 samples of clinical material were collected from the patients of the T. Marciniak Lower Silesian Specialist Hospital in Wrocław: blood (n = 5644. i.e., 41.7%), urine (n = 2782; 20.6%), wounds (n = 2133; 15.8%), RT (n = 1852; 13.7%), rectal swabs for CPE carrier status (n = 820; 6.1%), CSF (n = 175; 1.3%) and stool (n = 123; 0.9%) (Figure 11, Appendix A). A total of 3894 bacterial strains were isolated from the clinical material, which constitutes 28.8% of the total (3894/13528). During the microbiological identification, it was found that 965/3894 (24.8%) of the isolates belonged to the species *Escherichia coli (E.coli)*, 416/3894 (10.7%) belonged to *K. pneumoniae,* and 172/3894 (4.4%) belonged to *E. cloacae*, which in total constituted 1553/3894 (39.9%) of the isolates. Finally, 115/1553 (7.5%) *Enterobacteriaceae* CRE isolates (resistant to one or more carbapenem antibiotics) were isolated, including 58/115 (50.4%) *K. pneumoniae* isolates (Appendix A). In accordance with the guidelines (Table 1), 21 ETP-resistant *E. coli* isolates were isolated in the course of the study, including 2 ETP- and IMP-resistant and 36 *E. cloacae* isolates, including 32 ETP-resistant strains, 3-IMP and ETP-resistant strain and 1 strain resistent to ETP, MEM and IMP. Additionally, 58 ETP-resistant *K. pneumoniae* isolates were identified, including 30 resistant to all carbapenems tested, 7 to ETP and MEM, and one resistant to IMP only (Figure 1).

The largest number of CRE isolates came from departments I (23/115 (20.0%)) and XIV (18/115 (15.7%)). Most often, they were isolated from urine 41/115 (35.7%) and wounds 32/115 (27.8%). The numerical share of CRE isolates, taking into account the departments and type of material, is presented in Appendix A and in Figure 2 below.

### 2.1. Assessment of the Antibiotic Susceptibility of K. pneumoniae

Assessment of Drug Susceptibility to Meropenem and Imipenem

The MIC values of the studied *K. pneumoniae* isolates ranged from ≤0.25–<32 mg/L for meropenem and imipenem (Appendix A). 

Based on the determined MIC values for imipenem and meropenem of the studied *K. pneumoniae* isolates, a higher level of resistance was found in the CPE isolate group compared to the CRE isolates (Figure 3; Appendix A).

### 2.2. Susceptibility Assessment for Ertapenem

Using the disc diffusion method, 58/416 (13.9%) *K. pneumoniae* isolates showing resistance to ertapenem were detected. Out of the 58 strains, 16 (27.6%) were isolated from rectal swabs for CPE carrier status assessment (Figure 4; Appendix A). Then, 42 (72.4%) isolates of *K. pneumoniae* were further analysed. The resistant strains were characterized by zones of growth inhibition ranging from 6 mm to 24 mm.

### 2.3. Detection of Carbapenemases

*K. pneumoniae* that showed resistance to ETP and/or MEM were tested using phenotypic tests for the detection of KPC, MBL and OXA-48 carbapenemases. In the case of a positive (marked as +) or doubtful (marked as ±) result of the phenotypic test, immunochromatographic tests and the CarbaNP test were performed (Appendix A). CPE isolates were sent to the National Reference Centre for Antimicrobial Susceptibility to confirm resistance mechanisms by genetic methods.

In the case of CRE *K. pneumoniae*, carbapenemases were detected in 28/58 (48.3%) isolates (Appendix A). None of the tested isolates produced KPC-type cabapenemase. However, 23/28 *K. pneumoniae* isolates produced MBL/NDM (82.1%), 5/28 produced VIM (14.3%), and one produced MBL/NDM + OXA-48. Carbapenemases were detected in 13 out of 16 (81.3%) carbapenem-resistant *K. pneumoniae* isolates derived from rectal swabs for CPE carrier status assessment, including 11 NDM types and 2 VIM types (Figure 4; Appendix A).

In June 2020, a significant increase in the number of CPE isolates was recorded compared to the remaining months of the year. The tested CPE isolates belonged to the species *K. pneumoniae*, and all of them produced NDM-type carbapenemase (Figure 5).

Antibiograms were performed for 42 *K. pneumoniae* isolates, and the results are presented in Figure 6. More than 50% of the isolates were susceptible only to imipenem (59.5%) and gentamicin (57.1%), with concurrent resistance or reduced susceptibility to the other tested antibiotics. A great majority of the isolates tested showed resistance to piperacillin/tazobactam (90.5%), ceftazidime (90.5%), cefepime (88.1%) and ertapenem (100.0%).

From 42 strains of *K. pneumoniae*, 5 (11.9%) were isolated from RT, 5 (11.9%) from blood, 8 (18.1%) from wounds, and 24 (57.1%) from urine.

From *K. pneumoniae* strains isolated from RT, all proved to be resistant to ertapenem, and a proportion of the isolates showed reduced sensitivity to meropenem (20.0%) and amikacin (60.0%). All strains were sensitive to imipenem, and more than 80% of the strains were sensitive to meropenem and gentamicin (Figure 7).

*K. pneumoniae* isolates from wounds were mostly susceptible to imipenem (62.5%), gentamicin (62.5%) and meropenem (50.0%). However, all of these isolates were resistant to ertapenem and amoxicillin/clavulanic acid, and some of them showed reduced susceptibility to meropenem (12.5%), amikacin (62.5%) and ceftazidime (12.5%) (Figure 8).

*K. pneumoniae* isolated from urine often showed sensitivity to meropenem (58.3%) and imipenem (58.3%), while all were resistant to ertapenem and levofloxacin. Some isolates showed reduced susceptibility to amikacin (50.0%), imipenem (12.5%), meropenem (8.3%) and piperacillin/tazobactam (4.2%) (Figure 9).

Blood samples derived *K. pneumoniae* isolates were susceptible to colistin (40.0%), amikacin (20.0%) and imipenem (20.0%), while all were resistant to ertapenem, trimethoprim/sulfamethoxazole, ciprofloxacin, levofloxacin, cefepime, ceftazidime and piperacillin/tazobactam. Some isolates showed reduced susceptibility to meropenem (40.0%), imipenem (20.0%) and amikacin (20.0%) (Figure 10).

## 3. Discussion

Since the first detection of *K. pneumoniae*-produced carbapenemase in the USA in 2001, there has been a global increase in the number of reported cases of infections with bacteria that produce this resistance mechanism [17,18,19]. In Poland, the first case of the isolation of KPC carbapenemase was recorded in 2008, and in the same year, the presence of carbapenemases was confirmed in 31 samples obtained from patients hospitalized in five different hospitals in Warsaw, covering three outbreaks [20].

*Klebsiella* sticks producing carbapenemases were a serious problem in Poland from 2009 to 2012, when the epidemic clone ST258 dominated [21,22]. In the following years, the number of outbreaks caused by *Klebsiella* isolates producing NDM-1 enzymes increased significantly, and between 2016 and 2017, the epidemic clone ST11 was responsible for 98% of registered cases of infection. [5,19]. In recent years, hospital outbreaks caused by *Klebsiella* NDM-1 isolates have been reported practically all over Poland [23,24]. Currently, in the case of various Gram-negative rods, new variants of carbapenemases and epidemic clones spreading to different regions of the world are appearing more and more frequently [25,26].

As a result of growing concern surrounding the increase in AMR in bacteria, the WHO established a separate unit—the Global Antimicrobial Resistance Surveillance System (GLASS) [27]. Between 2015 and 2018, more than 2 million patients from 66 countries were considered in the extensive GLASS antimicrobial resistance studies. In some countries, the proportion of carbapenem-resistant isolates ranged from 30.0% to 60.0% [28]. Since 2015, following the example of GLASS, many other institutions monitoring the AMR phenomenon have been established around the world [29,30].

The vast majority of MBLs detected in Poland between 2006 and 2011 belonged to the VIM class (93.8%) [21,24]. In 2011, MBL in the NDM class was detected for the first time in Poland [5,19]. In recent years in Poland, *Klebsiella* KPC and NDM-1 isolates have been responsible for numerous epidemic outbreaks in hospitals [31,32,33].

Most CRE strains are CPE. An exception is for CRE with a carbapenem resistance mechanism other than carbapenemase production. On the other hand, CPE strains that show low MIC values of carbapenems and remain phenotypically sensitive to carbapenems are not included in CRE [34]. In the author’s own work, 1553 strains of the *Enterobacteriaceae* family were examined, which were isolated from 13,528 material samples from hospitalized patients (39.9% of all microbes grown). The tested isolates belonged to the species *K. pneumoniae*, *E. coli* and *E. cloacae*, of which 115 (7.5%) were classified as CRE. In the case of *K. pneumoniae* isolates, the rate of CPE was 28/58 (48.3%), while 30/58 (51.7%) were non-CPE isolates. A study by Lin et al. collected 128 CRE isolates from 128 patients: of which 69 (53.9%) were CPE and 59 (46.1%) were non-CPE. The majority of CPE isolates are KPC-2 (56.5%), NDM (39.1%), and IMP (5.8%). *K. pneumoniae* carbapenemase (KPC) clonal group 11 was the most common form of CPE. Additionally, it was found that antibiotic resistance was more common in the CPE group than in the non-CPE group [35]. A similar relationship was observed in the case of the drug susceptibility analysis of *K. pneumoniae* strains in our own research. Identification of resistance mechanisms for CPE/non-CPE and CRE is essential for better clinical management of patients.

The frequency of CRE strains from the *Enterobacteriaceae* family in our own study was comparable to that obtained in the study by Loqman et al. (7.5%) [36]. On the other hand, in studies conducted in Asia, the share of CRE strains was twice as high (16.5%) [37]. The results of our research were also compared with the reports of the centre monitoring the development of drug resistance on a European scale—the European Centre for Disease Prevention and Control (ECDC). Based on data containing information from 29 countries participating in the international EARS-Net project, it was found that between 2015 and 2019, *K. pneumoniae* strains resistant to imipenem and/or meropenem constituted 7.9% [38]. In our own study, the percentage of strains resistant to carbapenems was 13.9% for *K. pneumoniae*, which was consistent with the results for Europe and Poland published by ECDC.

The results of our own research indicate that over half of the tested CRE strains were *K. pneumoniae* bacteria (50.4%), while *E. cloacae* (31.3%) and *E. coli* (18.2%) were subsequently isolated. In the publications of Loqman et al. [36] and Ben Helal et al. [39], a similar trend was observed in the presence of CRE bacteria among *Enterobacteriaceae*. In the above-mentioned studies, the percentage of *K. pneumoniae* was 59.0% and 87.4%, respectively, 24.0% and 12.08% for *E. cloacae*, and 10.0% and 5.6% for *E. coli*. A different distribution was obtained in studies carried out in Asia and the Pacific, where the most common CRE strains were E. cloacae (46.1%), followed by *K. pneumoniae* (28.9%) and *E. coli* (20.2%) [37].

A disturbing phenomenon is still observed in the high colonization of patients’ digestive systems by CPE strains. Colonization, which is estimated outside hospitals at 5–38%, increases in a hospital environment, and in the case of hospitalized patients, the incidence in feces is as high as 77.0%, 19.0% in the nasopharynx, and 42.0% in the hands [6,16,40]. The emergence of CPE strains has increased awareness of the threat of epidemic outbreaks. Protective procedures include the screening of patients admitted to hospitals for CPE carrier status, tightening of antiseptic and disinfection procedures, and isolation of a colonized patient [41].

The materials from which the CRE strains were derived in our own study came mostly from departments dealing with the care of people in difficult clinical conditions, where the stay of a hospitalized person is statistically the longest. The most numerous group were the strains isolated from the Department of Anaesthesiology and Intensive Care (20.0%), Department of Traumatic Orthopedics (15.7%), Department of General Surgery (13.9%) and Department of Internal Medicine (13.0%). In the studies of Loqman et al., CRE strains were most often isolated from materials from the Neonatology Department (40.3%), Plastic Surgery Department (14.1%), then from the Urology and Nephrology Department, Children’s Emergency Department and the Intensive Care Department [36]. Differences in the quantitative distribution of the material in terms of origin may result from different characteristics of the work of health care institutions.

In our study, CRE isolates came from 5 groups of biological materials. The most numerous were materials from the urinary tract (35.7%) and wounds (27.8%). Materials from the RT, blood and rectal swabs for CPE carriers accounted for 12.2%, 9.6% and 13.9%, respectively. The results obtained are consistent with the studies by Falagas et al., in which CRE was most often isolated from urine [42]. As shown by the results obtained by other authors, the highest percentage of these bacteria in symptomatic infections was isolated from the lower respiratory tract and from urine cultures [24,35].

According to the results obtained in the course of this research, *K. pneumoniae* carbapenemase isolates constituted 48.3% (28/58) of the CRE group. Considering the results obtained using phenotypic, biochemical and genetic methods, as many as 23/28 (82.1%) of the isolates produced NDM-type carbapenemase, 5/28 (14.3%) produced VIM-type carbapenemase, and 1 simultaneously produced NDM and OXA-48. No isolates producing KPC and IMP carbapenemases were detected in the study. The CPE bacterial species most frequently isolated in the study was *K. pneumoniae*, with 58/115 (50.4%) isolates. The studies by Chmielewska et al., conducted in Poland, obtained similar results [43]. In the studies in the Mediterranean basin, the share of CPE/CRE isolates was 81.6% for the Tunisian study and 85.9% for the Moroccan study, respectively. In both studies, carbapenemase type OXA-48 and NDM were the most common isolates [36,39]. Based on the results of studies conducted in China, it was found that the most frequently isolated carbapenemase was the KPC type, and the NDM, IMP and OXA-48 types were less often isolated [44].

The carbapenemase-generating strains analysed in this study were isolated over the course of 2020 with a clearly uneven monthly distribution. In June 2020, a significant increase in the number of CPE isolates was recorded compared to the remaining months of the year. The tested CPE isolates belonged to the species *K. pneumoniae*, and all of them produced NDM-type carbapenemase.

According to the report concerning priority research on the development of new antibiotics published by the WHO in 2017, finding an alternative drug therapy for the treatment of infections caused by strains resistant to carbapenems is a priority, defined as critical [45]. The antibiotics currently used in cases of resistance to carbapenems are colistin and, in some cases, fosfomycin [46]. However, the constant reports on *Enterobacteriaceae* isolates, which are also resistant to the aforementioned antibiotics, are worrying. In a study conducted in Poland, a reduction in the sensitivity of *K. pneumoniae* MBL/NDM isolates to colistin was found from 100.0% in 2016 to 45.8% of strains susceptible in 2018, which significantly limits the therapeutic options in cases of infections caused by these microorganisms [47]. As shown by the results of our own research, in the case of CREs isolated from blood, the percentage of resistance to colistin was estimated at as much as 60.0% for *K. pneumoniae*. In the studies by Qadi et al. carried out in 2020, the proportion of CRE resistant to colistin was 31.6% for *K. pneumoniae*, which confirms the fact that modern medicine has limited possibilities of effective therapy in the treatment of infection caused by these microorganisms [48]. In cases of resistance to aminoglycosides with simultaneous sensitivity to colistin or tigecycline, a combination of colistin with tigecillin and meropenem can be used. Combined antibiotic therapy is an option for managing *K. pneumoniae* MBL/NDM infections [47,49,50,51,52].

The rapid evolution of drug resistance in *Klebsiella* also concerns new mechanisms determining resistance to aminoglycosides and quinolones (blaKPC and qnr on a common plasmid have already been described) [53,54]. In addition, mutations in the genes of the OmpK35 and OmpK36 porin proteins are responsible for blocking drug transport to the cell and an increase in the number of proton pumps in the cell membrane, which results in the removal of the antibiotic from the cell (AcrAB efflux pumps) [55,56]. Numerous studies confirm that KPC isolates may be sensitive to gentamicin; this gene is usually not on the same plasmid as the carbapenemase gene [57,58]. The situation is different in the case of plasmids encoding NDM-1, which also carry genes for aminoglycoside resistance [59]. As shown by the results of our own studies, in the case of CRE, the percentage of resistance to gentamicin was estimated at 42.9% for all tested *K. pneumoniae*, with blood-derived isolates, which mostly presented the NDM-1 phenotype, 100.0% resistant to gentamicin, ciprofloxacin and levofloxacin. On the other hand, some isolates resistant to colistin and carbapenems are susceptible to fosfomycin. However, the percentage of resistance to this antibiotic in the case of carbapenemase-producing isolates is high, which indicates the relationship of fosfomycin resistance genes with plasmids encoding carbapenemases [46].

*K. pneumoniae* CPE producing MBL/NDM-type carbapenemases are classified as bacteria with effective mechanisms of resistance to antibacterial drugs, which is a significant threat to public health [2,7]. In order to prevent the spread of CPE strains, screening tests should be introduced in patients admitted to hospital wards, irrespective of their previous medical history, as CRE and CPE isolates are increasingly being observed in outpatients. In the hospital, which is the subject of the presented study, only 820 such screening tests were performed during 12 months, while the number of patients admitted during this period was considerably higher. Early detection and isolation of a CPE-carrier patient in combination with rational antibiotic therapy could exclude the transmission of this pathogen and thus prevent the occurrence of epidemic outbreaks [15,60].

## 4. Materials and Methods

### 4.1. Bacterial Strains

A total of 13,528 clinical samples of material collected from patients of the T. Marciniak Lower Silesian Specialist Hospital in Wroclaw were analysed in the period from 1 January 2020 to 31 December 2020. 

The clinical materials tested were described as: blood (n = 5644), urine (n = 2782), wound secretions (n = 2133), RT—respiratory secretions (n = 1852), rectal swabs on CPE carriers (n = 820), CSF—cerebrospinal fluid (n = 174) and stool (n = 123). Figure 11 presents the proportions of the analysed clinical materials.

The clinical materials studied came from patients hospitalized in 15 departments of T. Marciniak Lower Silesian Specialist Hospital in Wroclaw. In accordance with the percentage share, the most studied materials were collected from patients of the Department of Internal Medicine (20.4%) and the Department of Anaesthesiology and Intensive Care (17.5%) (Appendix A). In Appendix A and Figure 12, the numerical share of clinical materials provided, broken down by research ordering units, are presented.

### 4.2. Microbiological Assays

The studies were carried out in accordance with the outbreak treatment scheme: from drug resistance analysis to the detection of carbapenemases by enzymatic and disc diffusion methods [15]. To assess the sensitivity of selected *Enterobacteriaceae* to carbapenems, the guidelines published by EUCAST for 2020 were followed [17,18]. Bacterial strains with MIC values and zones of growth inhibition exceeding the established breakpoints for MEM, ETP and IMP were considered resistant (marked with the letter R) (Table 1). 

#### 4.2.1. Automated System

The VITEK2 automated system (bio Mérieux) was used to identify and evaluate the bacterial susceptibility to drugs. Identification cards were used to identify Gram-negative fermenting and non-fermenting bacteria. Identification was based on 47 biochemical tests, enzyme activity and drug resistance tests. Two types of antibiogram cards for Gram-negative bacteria were used to assess drug susceptibility with the VITEK2 system. These were the VITEK2 AST-331 cards (amikacin, ampicillin/sulbactam, aztreonam, cefepime, ceftazidime, ciprofloxacin, colistin, gentamicin, imipenem, levofloxacin, meropenem, piperacillin, piperacillin/tazobactam, ticarcillin/clavulanic acid, tobramycin, trimethoprim/sulfa) and VITEK2 AST-332 (amikacin, amoxicillin/clavulanic acid, cefepime, cefotaxime, ceftazidime, cefuroxime, ciprofloxacin, colistin, ESBL confirm, gentamicin, imipenem, meropenem, piperacillin/tazobactam, tygecycline, tobramycin, trimethoprim/sulfa) set of 16 antibiotics. Using this system made it possible to determine the value of MIC. 

#### 4.2.2. Disc Diffusion Method

Drug susceptibility was assessed by the disc diffusion method according to EUCAST recommendations [61,62]. For this purpose, several colonies of similar morphology were suspended in 0.9% NaCl solution with a sterile swab until an inoculum with a density of 0.5 McFarland was obtained. The density of the suspension was measured with a DensiCheck plus photometric device (Biomerieux, Crappone, France). Then, within 15 min of preparation of the inoculum, the prepared suspension was inoculated onto the MHA using a new, sterile cotton swab. Two antibiotic discs, ertapenem 10 µg (ETP10) and temocillin 30 µg (TEM30), were applied within 15 min of inoculation and incubated at 35 ± 1 °C for 18 ± 2 h. Performing an additional disc diffusion test was necessary due to the inability to determine MIC for ETP and TEM using VITEK2 AST cards.

#### 4.2.3. Detection of Carbapenemases

According to the EUCAST guidelines for the detection of carbapenemases, it is recommended to test all *Enterobacteriaceae* with reduced sensitivity to ertapenem or meropenem [61,62]. In these studies, the values in Table 1 were used as susceptibility breakpoints.

At a later stage of the study, a combination of phenotypic tests was used to identify CPE. Combined disc methods, cassette immunochromatographic tests and colourimetric tests based on carbapenem hydrolysis were used. For the detection of carbapenemases in the KPC class, the disc diffusion method was used in accordance with the EUCAST method [64]. Discs with meropenem 10 µg (MEM10) and meropenem 10 µg + phenylboronic acid (MEM10 + KB) (GRASO BIOTECH, Starogard Gdański, Poland) were used. Isolates in which the difference in the size of the growth inhibition around the MEM10 disc was 4 mm or greater than that around the MEM10 + KB disc were considered to be KPC-positive. The disc diffusion method, according to the EUCAST methodology, was used to detect MBL class carbapenemases [20]. EDTA discs with 30 µg ceftazidime (CAZ30) and 10 µg imipenem (IMP10) were used. When reading the test results, significant enlargement of the zone of inhibition of growth around the CAZ30 and/or IMP10 disc on the side of the EDTA-soaked disc was interpreted as positive. 

Detection of OXA-48 carbapenemases was performed using the temocillin 30 µg disc test (TEM30) [64,65]. The test was routinely performed on all isolates for which drug susceptibility was assessed using the disc diffusion method. A reading indicating a reduction in diameter of the growth inhibition around the TEM30 disc of less than or equal to 10 mm was taken as a positive test result.

A rapid biochemical test based on the hydrolysis of imipenem by carbapenemases released from the lysate of bacterial cells was used to detect carbapenemases in *Enterobacteriaceae.* Hydrolysis of the IMP leads to a decrease in the pH of the reaction medium, and as a consequence, a change in the colour of phenol red from red to yellow or orange is observed. A positive result of the Carba NP test in the case of *Enterobacteriaceae* confirms with 100% certainty the presence of carbapenemases in the tested isolate. The test was performed in accordance with the KORLD recommendations [66].

Resist-5 O.O.K.N.V. immunochromatographic tests (Argenta sp.z o.o., Poznań, Poland) were used to identify carbapenemases detected in *Enterobacteriaceae*. On the basis of the tests used, it was possible to identify the strains producing the enzymes KPC, OXA-163, OXA 48, NDM and VIM.

The multiplex PCR method was used to identify the genes encoding the NDM, VIM and KPC carbapenemases. For this purpose, appropriately designed primers NDM-F, NDM-R, VIM-UF, VIM-UR, KPC-A and KPC-B, as well as OXA-48a and OXA-48b, were used. The sequences of the primers used and the list of genes for which they are dedicated are presented in Table 2 [66]. Primers marked with a 1 were created in KORLD. 

In order to perform the tests, the Veriti 96-well Thermal Cycler from Applied Biosystems (Applied Biosystems, Waltham, MA, USA) and the T100 Thermal Cycler from BioRad (BioRad, Hercules, CA, USA) were used.

## 5. Conclusions

In June 2020, a significant increase in the number of *K. pneumoniae* isolates producing the same type of carbapenemases (NDM) was found.

A significant rate in infections of CRE and CPE isolates proves the necessity to test patients admitted to hospital wards for CPE carrier status in order to limit the emergence of new epidemic outbreaks.

## Figures and Tables

**Figure 1 pathogens-11-00859-f001:**
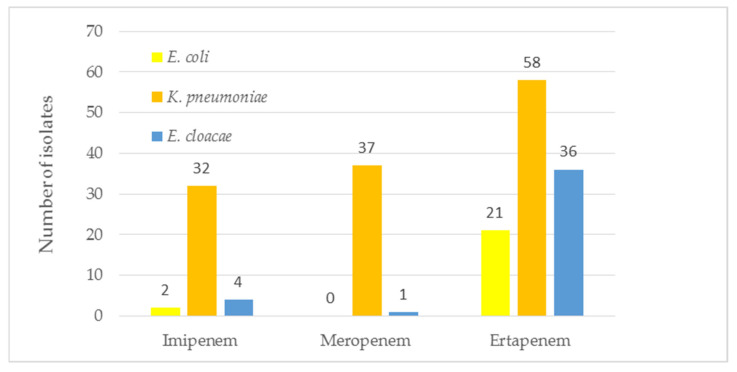
The number of carbapenem-resistant isolates of *Enterobacteriaceae*.

**Figure 2 pathogens-11-00859-f002:**
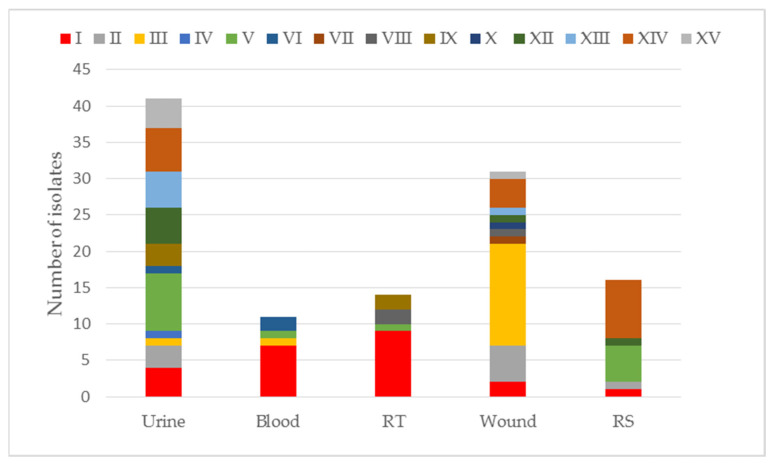
The number of CRE isolates, taking into account the hospital departments and the type of material. RS—rectal swabs for CPE carrier status; RT—respiratory tract; I—Department of Anaesthesiology and Intensive Care, II—Department of Pediatric Surgery, III—Department of General and Vascular Surgery, IV—Department of Plastic Surgery, V—Department of Internal Medicine, VI—Department of Endocrinology, Diabetology and Internal Medicine, VII—Department of Cardiology, VIII—Department of Neurosurgery, IX—Department of Neurology with the Stroke Division, X—Department of Rheumatology and Internal Medicine, XI—Department of Rehabilitation, XII—Department of Hospital Emergency, XIII—Department of Toxicology and Internal Medicine, XIV—Department of Orthopedic Surgery, XV—Department of Urology and Urological Oncology.

**Figure 3 pathogens-11-00859-f003:**
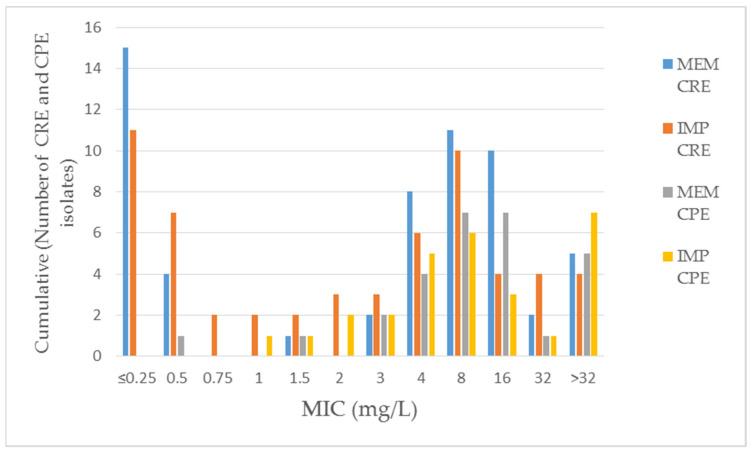
Cumulative MIC of CPE and CRE *K. pneumoniae* against carbapenems. RS—rectal swabs for CPE carrier status; RT—respiratory tract.

**Figure 4 pathogens-11-00859-f004:**
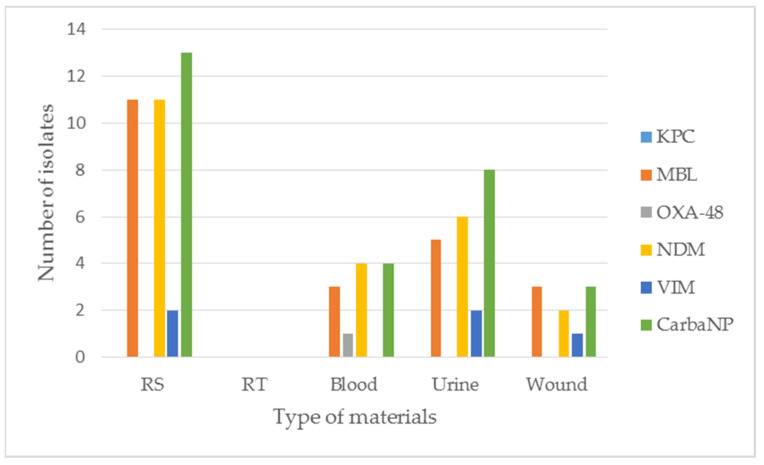
The results of phenotypic, immunochromatographic and CarbaNP tests used to detect carbapenemases in studied *K. pneumoniae* isolates in 2020. RS—rectal swabs for CPE carrier status; RT—respiratory tract.

**Figure 5 pathogens-11-00859-f005:**
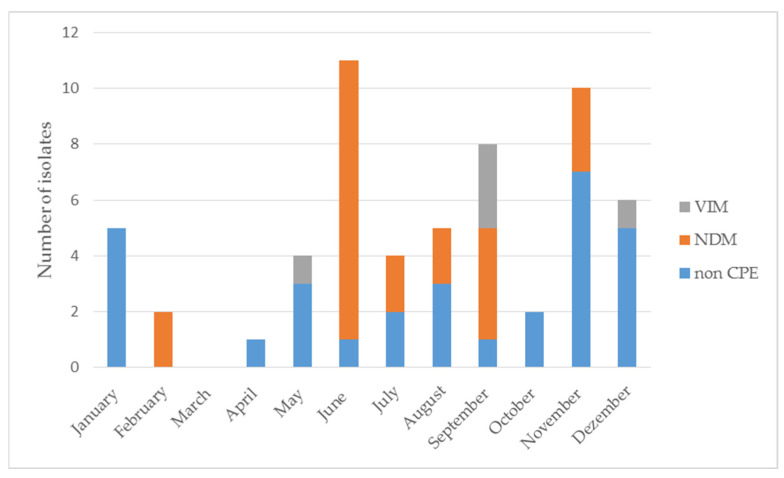
The number of CRE isolates by month.

**Figure 6 pathogens-11-00859-f006:**
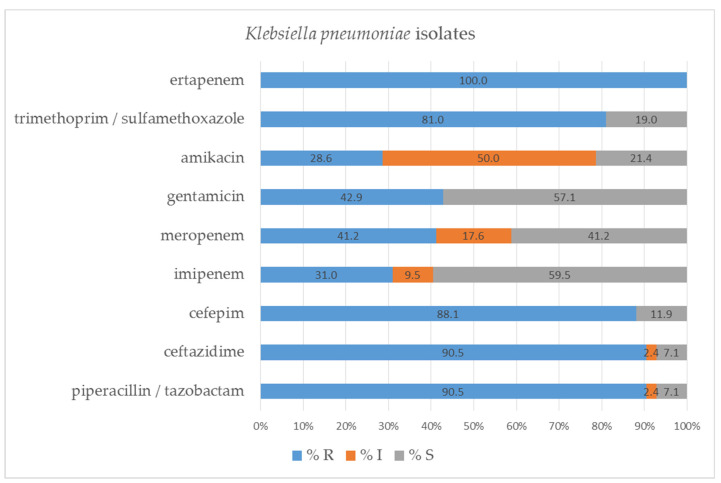
Percentage of tested *Klebsiella pneumoniae* isolates susceptible and resistant to the selected antibiotics, determined by VITEK-2 automated system; S—susceptible; I—susceptible, increased exposure; R—resistant.

**Figure 7 pathogens-11-00859-f007:**
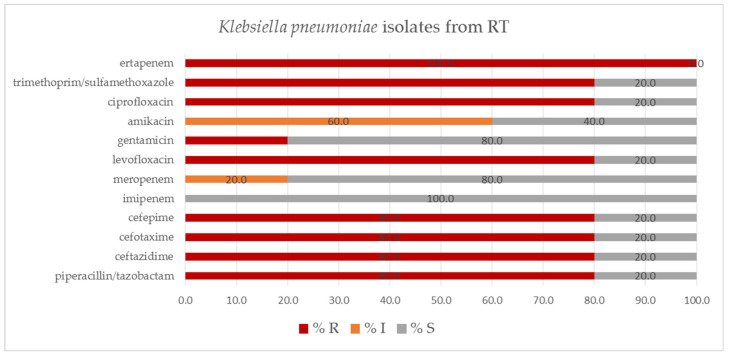
Percentage of tested *Klebsiella pneumoniae* isolates from RT susceptible and resistant to the selected antibiotics, determined by VITEK-2 automated system; S—susceptible; I—susceptible, increased exposure; R—resistant; RT—respiratory tract.

**Figure 8 pathogens-11-00859-f008:**
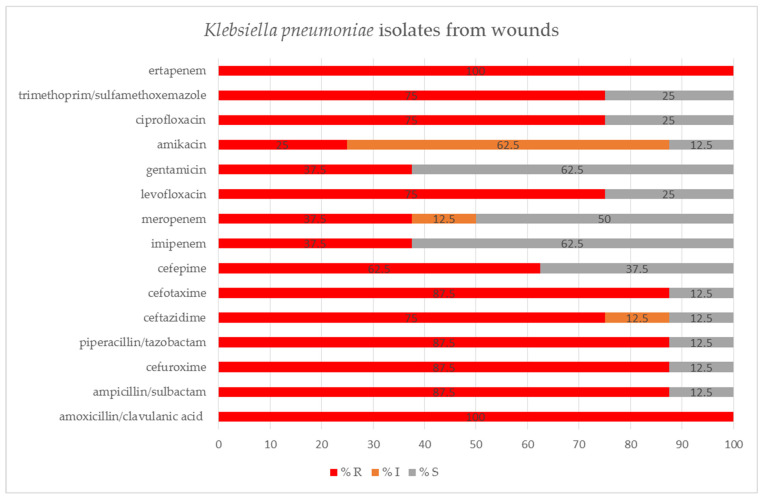
Percentage of tested *Klebsiella pneumoniae* isolates from wounds susceptible and resistant to the selected antibiotics, determined by VITEK-2 automated system; S—susceptible; I—susceptible, increased exposure; R—resistant.

**Figure 9 pathogens-11-00859-f009:**
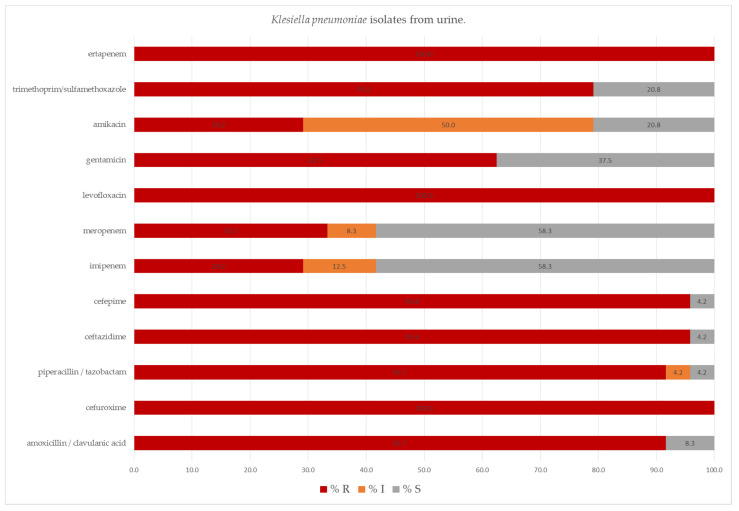
Percentage of tested *Klebsiella pneumoniae* isolates from urine susceptible and resistant to the selected antibiotics, determined by VITEK-2 automated system; S—susceptible; I—susceptible, increased exposure; R—resistant.

**Figure 10 pathogens-11-00859-f010:**
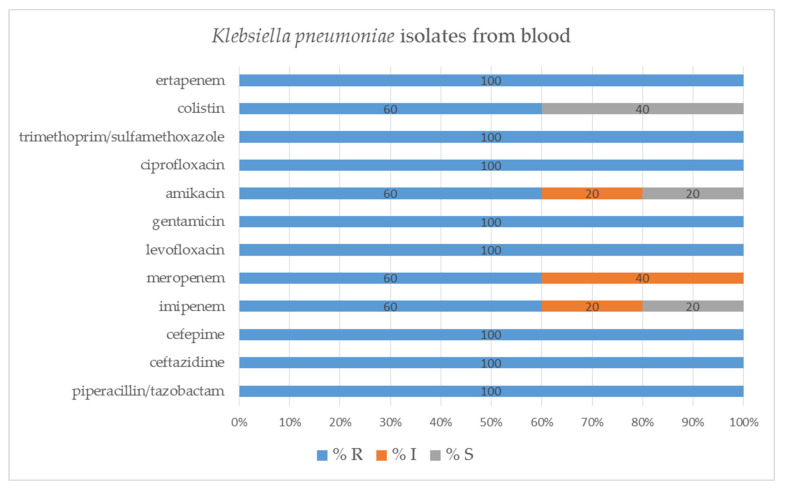
Percentage of tested *Klebsiella pneumoniae* isolates from blood susceptible and resistant to the selected antibiotics, determined by VITEK-2 automated system; S—susceptible; I—susceptible, increased exposure; R—resistant.

**Figure 11 pathogens-11-00859-f011:**
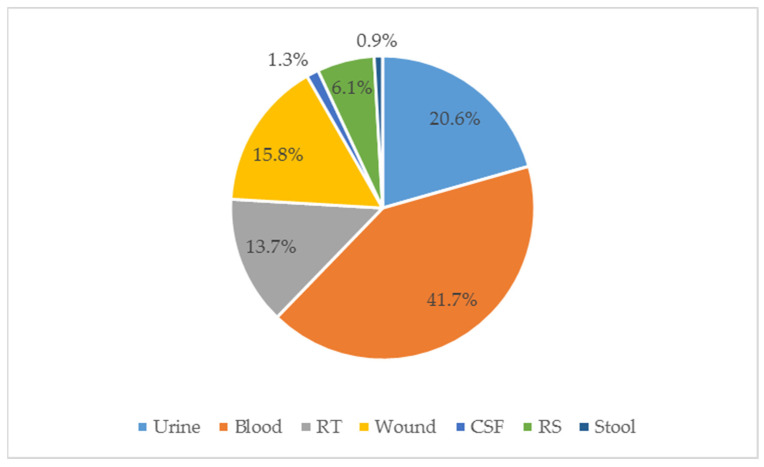
Type of biological material isolated from patients of a hospital in Wroclaw in 2020. RS—rectal swabs for CPE carrier; RT—respiratory tract; CSF—cerebrospinal fluid.

**Figure 12 pathogens-11-00859-f012:**
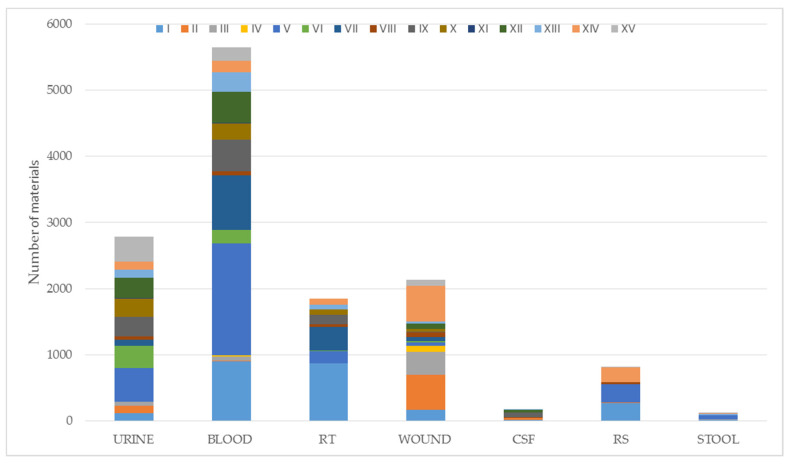
The number of analyzed clinical materials in 2020, taking into account the ordering department and the type of material. RS—rectal swabs for CPE carrier status; RT—respiratory tract; CSF—cerebrospinal fluid; I—Department of Anaesthesiology and Intensive Care, II—Department of Pediatric Surgery, III—Department of General and Vascular Surgery, IV—Department of Plastic Surgery, V—Department of Internal Medicine, VI—Department of Endocrinology, Diabetology and Internal Medicine, VII—Department of Cardiology, VIII—Department of Neurosurgery, IX—Department of Neurology with the Stroke Division, X—Department of Rheumatology and Internal Medicine, XI—Department of Rehabilitation, XII—Department of Hospital Emergency, XIII—Department of Toxicology and Internal Medicine, XIV—Department of Orthopedic Surgery, XV—Department of Urology and Urological Oncology.

**Table 1 pathogens-11-00859-t001:** Clinical breakpoints for meropenem, ertapenem and imipenem *for Enterobacteriaceae* [61,62,63].

	MIC Value (mg/L)		The Size of the Zone Growth Inhibition, Disc 10 µg (mm)	
Antibiotic	S≤	R>	S≤	R>
Meropenem	2	8	22	16
Ertapenem	0.5	0.5	25	25
Imipenem	2	4	50	17

MIC: Minimal Inhibitory Concentration.

**Table 2 pathogens-11-00859-t002:** List of primers used by KORLD for the detection of carbapenemases and the identification of bacteria by PCR methods [66].

Name of Starter	Gen	Starter Sequence
OXA-48a	*bla_OXA−_* _48_	5-TTGGTGGCATCGATTATCGG-3
OXA-48b	*bla_OXA−_* _48_	5-GAGCACTTCTTTTGTGATGGC-3
KPC-A	*bla_KP C−_* _2_	5-CTGTCTTGTCTCTCATGGCC-3
KPC-B	*bla_KP C−_* _2_	5-CCTCGCTGTGCTTGTCATCC-3
khe-F	*khe*	5-ACCATGTCCGATTTAATCACAACACGC-3
khe-R	*khe*	5-GCAGACGAACTTCCTGCTCGGT-3
VIM-UF	*bla_V IM_*, *bla_IMP_*	5-GTTTGGTCGCATATCGCAAC-3
VIM-UR	*bla_V IM_*	5-TCAATCTCCGCGAGAAG-3
NDM-F^1^	*bla_NDM_*	5-CAGCACACTTCCTATCTCGAC-3
NDM-R^1^	*bla_NDM_*	5-GTAGTGCTCAGTGTCGGCATC-3

## Data Availability

Not applicable.

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
