# Peer review of "Occurrence and Characteristics of Carbapenem-Resistant *Klebsiella pneumoniae* Strains Isolated from Hospitalized Patients in Poland—A Single Centre Study"

_pathogens, 2022, doi:10.3390/pathogens11080859_

Round 1

Reviewer 1 Report

1.          The data in this study didn’t provide too much new information. The results were scattered and need to reorganize.

2.          Bacteria name should be in italics.

3.          Please check the format of abstract. In current format, the “introduction:”, “Material and methods:”, “Results:” and “Conclusions:” should be deleted.

4.          ESBL, K. pneumoniae, ETP, MEM, MBL, CPE ang CRE should be spell out for the first time in the abstract and manuscript.

5.          For Fig. 3 and 4, The number of isolates of different MIC are not continuous, it’s inappropriate to show with line charts. I suggest to use grouped bar chart to combine two figures and it’s easier to compare.

6.          For Fig. 5, the data is about number of isolates, so the chart should not contain error bars.

Author Response

Response to Reviewer 1 comments

Dear Reviewer,

Thank you very much for reviewing our manuscript. 

Below I provided a point-by-point response to the Reviewer’s comments.

1.The data in this study didn’t provide too much new information. The results were scattered and need to reorganize.

Response 1: As suggested by the Reviewer, the form of presenting the results was changed. A graph has been added and the data in Figures 3 and 4 have been placed in one Figure (Figure 3).

  1. Bacteria name should be in italics.

Response 2: Corrected as suggested by the Reviewer.

  1. Please check the format of abstract. In current format, the “introduction:”, “Material and methods:”, “Results:” and “Conclusions:” should be deleted.

Response 3: It has been removed.

  1. ESBL, K. pneumoniae, ETP, MEM, MBL, CPE ang CRE should be spell out for the first time in the abstract and manuscript.

Response 4: Corrected as suggested by the Reviewer.

  1. For Fig. 3 and 4, the number of isolates of different MIC are not continuous, it’s inappropriate to show with line charts. I suggest to use grouped bar chart to combine two figures and it’s easier to compare.

Response 5: Figures 3 and 4 have been transferred to the supplement. A new Figure (Figure 3) was inserted in the manuscript as suggested by the Reviewer.

  1. For Fig. 5, the data is about number of isolates, so the chart should not contain error bars.

Response 6: Error bars have been removed.

Reviewer 2 Report

It is an important study that characterize the percentage of carbapenemases producing  Klebsellia pneumoniae in Poland. The authors assessed this point using a large number of cohort , though it is derived from a single center.   The authors analyzed blood, urine, and rectal swab, and they reported that some K.pneumoniae isolates resistant to 1 or more carbapenem antibiotics, and 27.6% were isolated from rectal  swabs carbapenemases were detected in  48.3% of isolates. K. pneumoniae isolates produced MBL / NDM (82.1%), VIM (14.3%), MBL/NDM + OXA-48. Carbapenemases were detected in 81.3% carbapenem- resistant K. pneumoniae isolates derived from rectal swabs.

Major comments

1- IRB no of the study is not reported

2- There is lack of representative image of combined test assay for a carbapenemase detection and also the molecular assay.

3- To determine MIC for carbapenemases producing isolates, it is highly recommended to do E-test.

4- The correlation between phenotypic and genotypic detection of carbapenemases should be determined

Minor comments

The title need to be rephrased 

Author Response

Response to Reviewer 2 comments

Dear Reviewer,

Thank you very much for reviewing our manuscript. 

Below I provided a point-by-point response to the Reviewer’s comments.

  1. IRB no of the study is not reported

Response 1: The tests were performed in a diagnostic laboratory.

  1. There is lack of representative image of combined test assay for a carbapenemase detection and also the molecular assay.

Response 2: The required molecular analyzes to confirm the presence of carbapenemases were performed in KORLD (National Reference Center for Antimicrobial Susceptibility) using the VITEK2 automated system (bio Mérieux) and using identification cards. Unfortunately, we do not have detailed data on the tests performed.

  1. To determine MIC for carbapenemases producing isolates, it is highly recommended to do E-test.

Response 3: The tests were carried out in a diagnostic laboratory, where the VITEK2 automated system (bio Mérieux) and using identification cards were used to determine the MIC value, in accordance with the laboratory diagnostic procedure and the recommendations of KORLD and the POLMICRO-  Central Microbiological Research and Quality Center.

  1. The correlation between phenotypic and genotypic detection of carbapenemases should be determined

Response 4: Our study concerns the analysis of the results obtained from the laboratory. We did not have the strains and therefore we could not perform additional analyzes

  1. Minor comments. The title need to be rephrased.

Response 5: The title has been rephrased.

Round 2

Reviewer 2 Report

Although the authors did not provide sufficient information about the test done in the study, the study is worthy to be published especially for residents in Poland.

I would suggest the authors to know the details for every test done for future publications